

**Soil infiltration characteristics and pore distribution under**
**freezing-thawing conditions**
**Ruiqi Jiang[1,2,3,★], Tianxiao Li[1,2,3,★], Dong Liu[1,2,3,*], Qiang Fu [1,2,3,*], Renjie Hou [1,2,3],**
**Qinglin Li [1], Song Cui [1,2,3], Mo Li [1,2,3]**
[1] School of Water Conservancy & Civil Engineering, Northeast Agricultural University, Harbin 150030,
China
[2] Key Laboratory of Effective Utilization of Agricultural Water Resources of Ministry of Agriculture,
Northeast Agricultural University, Harbin, Heilongjiang 150030, China
[3] Heilongjiang Provincial Key Laboratory of Water Resources and Water Conservancy Engineering in Cold
Region, Northeast Agricultural University, Harbin, Heilongjiang 150030, China
**\* Dong Liu and Qiang Fu are corresponding authors.**
**★ These authors contributed equally to this work.**
*Corresponding author at: School of Water Conservancy and Civil Engineering, Northeast Agricultural
University, Harbin, Heilongjiang 150030, China
*Correspondence to*: liudong9599@yeah.net (Dong Liu). fuqiang0629@126.com (Qiang Fu)
**Abstract.** Frozen soil infiltration widely occurs in hydrological processes such as seasonal soil freezing and
thawing, snowmelt infiltration, and runoff. Accurate measurement and simulation of parameters related to
frozen soil infiltration processes are highly important for agricultural water management, environmental
issues and engineering problems in cold regions. Temperature changes cause soil pore size distribution
variations and consequently dynamic infiltration capacity changes during different freeze-thaw periods. To
better understand these complex processes and to reveal the freeze-thaw action effects on soil pore





distribution and infiltration capacity, selected black and meadow soils and chernozem, which account for the
largest arable land area in Heilongjiang Province, China. Laboratory tests of soils at different temperatures
were conducted using a tension infiltrometer and ethylene glycol aqueous solution. The stable infiltration
rate, hydraulic conductivity were measured, and the soil pore distribution was calculated. The results
indicated that for the different soil types, macropores, which constituted approximately 0.1% to 0.2% of the
soil volume under unfrozen conditions, contributed approximately 50% of the saturated flow, and after soil
freezing, the soil macropore proportion decreased to 0.05% to 0.1%, while their saturated flow proportion
decreased to approximately 30%. Soil moisture froze into ice crystals inside relatively large pores, resulting
in numerous smaller-sized pores, which reduced the number of macropores while increasing the number of
smaller-sized mesopores, so that the frozen soil infiltration capacity was no longer solely dependent on the
macropores. After the ice crystals had melted, more pores were formed within the soil, enhancing the soil
permeability.
**Key words:** Freezing-thawing soil; Hydraulic conductivity; Pore distribution; Macropores; Infiltration
characteristics

## 37    1 Introduction

Over the last few decades, the temperature changes caused by global warming have altered the freezing state
of near-surface soils, and in China, changes in characteristic values such as the mean annual area extent of
seasonal soil freeze/thaw state and maximum freezing depth indicate the degradation of frozen soil,
especially at high latitudes (Wang et al., 2019;Peng et al., 2016). Under the effect of temperature, most frozen
regions experience the seasonal freezing and thawing of soil, accompanied by coupled soil water and heat
movement and frost heave processes, thus making the soil structure and function more variable (Oztas and
Fayetorbay, 2003;Fu et al., 2019;Gao et al., 2018). Parameters such as the soil infiltration rate and hydraulic



conductivity are key factors in the study of soil water movement, groundwater recharge, and solute and
contaminant transport simulation (Angulo-Jaramillo et al., 2000). In regard to unfrozen soils, the temperature
has been shown to change the soil structure and kinematic viscosity of soil water, thereby affecting the
unsaturated hydraulic conductivity of soils (Gao and Shao, 2015). In terms of frozen soils, the water
infiltration characteristics and pore size distribution are highly variable and difficult to observe (Watanabe
et al., 2013); moreover, the water movement in freezing-thawing soils is complicated by the migration of
water and heat and the associated water phase change (Jarvis et al., 2016). The accurate measurement of
water movement parameters and soil pore distribution under freeze-thaw conditions is a necessary
prerequisite for the quantitative description of the water movement in frozen soil, and the mechanism and
degree of influence of the temperature on the infiltration rate, hydraulic conductivity, porosity and other
parameters in the different stages of freeze-thaw periods require further research.
Currently, the studies related to the quantitative characterization of freezing-thawing soil infiltration can be
mainly divided into experimental and model studies. Field experiments have been performed less often
because under natural conditions, the infiltration water establishes a preferential flow into the deep soil, and
the alternating freeze-thaw effect forms ice crystals to block the flow path through large pores, subsequently
limiting water infiltration (Daniel et al., 1997), while the melting effect of the infiltration water on ice makes
it difficult to reach a steady infiltration state. Controlled laboratory experiments provide new opportunities
for the simulation of frozen soil infiltration processes and the measurement of infiltration parameters.
Williams and Burt (1974) conducted early direct measurements in the laboratory, resolved the water freezing
problem by adding lactose and applied dialysis membranes on both sides of soil columns, and they
determined the water conductivity of saturated specimens in the horizontal direction (Burt and Williams,
1976). Andersland et al. (1996) measured the hydraulic conductivity of frozen granular soils at different



saturations using a conventional drop permeameter with decane as the permeant and concluded that the
hydraulic conductivity was the same as that of unfrozen soils with water as the infiltration solution.
McCauley et al. (2002) determined and compared the differences in hydraulic conductivity, permeability and
infiltration rate between frozen and unfrozen soils using diesel mixtures as permeants, and their results
indicated that the ice content determines whether soil is sufficiently impermeable. Zhao et al. (2013)
quantified the unsaturated hydraulic conductivity of frozen soil using antifreeze instead of water, adopted a
multistage outflow method under controlled pressures and introduced the pore impedance coefficient.
However, most of these studies did not consider the differences in kinematic viscosity and surface tension
between soil water and other solutions, which often results in hydraulic conductivity estimation, and the
homemade devices in the laboratory are often inconvenient for generalization in the field. Due to the dynamic
changes in the temperature and moisture phase, direct measurement is difficult, and hydraulic conductivity
empirical equations and models of frozen soil have been developed. First, the frozen soil hydraulic
conductivity was simply considered to follow a power exponential relationship with the temperature (Nixon,
1991;Smith, 1985), while others considered the hydraulic conductivity of frozen soil to be equal to that of
unfrozen soil at the same water content and assumed that the hydraulic conductivity of frozen soil was a
function of the moisture content of unfrozen soil (Lundin, 1990;Flerchinger and Saxton, 1989;Harlan, 1973).
On the basis of Campbell's model (Campbell, 1985), Tarnawski and Wagner (1996) proposed a frozen soil
hydraulic conductivity model based on the soil particle size distribution and porosity. Watanabe and Wake
(2008) viewed soil pores as cylindrical capillaries and suggested that ice formation occurs at the center of
these capillaries and established a model to describe the movement of thin film water and capillary water in
frozen soil based on the theory of capillaries and surface absorption (Watanabe and Flury, 2008). The
similarity between freezing and soil moisture profiles has been demonstrated (Spaans and Baker,



1996;Spaans, 1994), and subsequently, freezing profiles have been applied to estimate the unsaturated
hydraulic conductivity of frozen soils (Azmatch et al., 2012), which has been combined with field tests and
inversion models to achieve a high accuracy (Cheng et al., 2019).
Understanding the distribution characteristics of the soil pore system is essential for the evaluation of the
water and heat movement processes in soil. The soil macroporosity has been shown to impose a major impact
on water cycle processes such as infiltration, nutrient movement and surface runoff. (Demand et al.,
2019;Jarvis, 2007). The macroporosity is widespread in a variety of soils and produces preferential flow in
both frozen and unfrozen soils (Mohammed et al., 2018;Beven and Germann, 2013), and the prefreeze
moisture conditions affect the amount and state of ice in the macropores of frozen soils, resulting in a notable
variability in the infiltration capacity of thawed soils (Hayashi et al., 2003;Granger et al., 1984). Field
experiments on frozen soil have also demonstrated that macropores accelerate the infiltration rate (Stähli et
al., 2004;Kamp et al., 2003), the number and size of macropores affect the freezing and infiltration capacity
of soil layers to different extents, and low temperatures cause infiltration water to refreeze inside macropores
(Watanabe and Kugisaki, 2017;Stadler et al., 2000). Research on the frozen soil macroporosity has largely
focused on the qualitative analysis of its impact on the soil structure and infiltration capacity, and with the
development of experimental techniques, certain new methods and techniques, such as computed
tomography (CT) and X-ray scanning, have been applied to measure the number and distribution of
macropores (Taina et al., 2013;Bodhinayake et al., 2004;Grevers et al., 1989), but the lack of sampling
techniques targeting frozen soil still restricts related research.
Many limitations and deficiencies remain in the direct measurement of frozen soil infiltration characteristics
and pore distribution, and the relevant models also require a large amount of measured data to meet the
accuracy and applicability requirements. In this paper, the stable infiltration rate and hydraulic conductivity



of three types of soils at different temperatures were measured by precise control of the soil and ambient
temperatures, and the macropore and mesopore size distribution was calculated by using a tension
infiltrometer and a glycol aqueous solution as the infiltration medium. The conclusions provide a basis and
reference for the numerical simulation of the coupled water-heat migration process of freezing-thawing soil
and related parameterization studies.
**2 Materials and methods**
**2.1 Test plan**
Referring to arable land area data of various regions of Heilongjiang Province, the three types of soils that
dominate the cultivated land area in Heilongjiang Province are black and meadow soils and chernozem
(Bureau, 1992). Harbin, Zhaoyuan and Zhaozhou were selected as typical soil areas for sampling. A 5-cm
surface layer of floating soil and leaves was removed, and topsoil samples were collected at depths ranging
from 0-20 cm. After natural air drying and artificial crushing, the soil was sieved, and particles larger than 2
mm in diameter were removed. The remainder was used to prepare soil columns. The basic physical and
chemical parameters of the test soils, such as the bulk density, organic content and mechanical parameters,
are listed in Table 1.
**Table 1**
**Basic physical and chemical properties of three kind of soils**

| Soil types | Bulk density (g/cm$^3$) | Organic content (g/kg) | Electrical conductivity (s/m) | Particle size (sand-silt-clay) (%) | Soil texture |
|---|---|---|---|---|---|
| Black soil | 1.31 | 28.32 | 0.02 | 12.64-70.82-16.54 | silt loam |



| | | | | |
|---|---|---|---|---|
| Meadow soil | 1.22 | 16.51 | 0.01 | 9.52-73.00-17.48 |
| Chernozem | 1.15 | 26.52 | 0.01 | 38.99-50.30-10.71 |

An artificial climate chamber was applied to control the temperature of the soil column and infiltration
solution, and four temperature treatments were established with three replications for each treatment: 15°C,
unfrozen soil, representing the soil before freezing, which was recorded as 15°C (BF); -5°C, stable freezing;
-10°C, stable freezing; and freezing at -10°C followed by thawing at 15°C, representing the soil after melting,
which was recorded as 15°C (AM). The freezing and thawing times were both 48 h. When the soil
temperature was consistent with the set temperature in the climate chamber, the samples were considered to
be completely frozen, and the effect of the number of freezing and thawing cycles was not considered in this
test. According to the basic information of the original soil, the volumetric moisture content of the sieved
soil was 30%, with a dry bulk density of 1.2 g/cm$^3$. To ensure a homogeneous column, the soil was loaded
into a polyvinyl chloride (PVC) cylinder at 5-cm depth intervals, and petroleum jelly was applied to the sides
to reduce the sidewall flow (Lewis and Sjöstrom, 2010). The PVC cylinder was 26 cm in diameter and 30
cm in height, with a perforated plate at the bottom. To prevent lateral seepage, the barrel occurred 5 cm
above the soil surface, and the thickness of the soil layer was 20 cm. A HYDRA-PROBE II sensor
(STEVENS Water Monitoring Systems, Inc., Portland, Oregon, USA) was inserted in the middle of the pail
to observe the potential soil temperature and liquid water content change to determine whether ice melting
occurred. A 5-cm thick layer of sand and gravel was emplaced below the soil column, and a 5-cm thick layer
of black polypropylene insulation cotton was wrapped around the outer layer and bottom of the soil column.
The stable infiltration rate under tension levels of -3, -5, -7, -9, -11, and -13 cm was measured with a tension





infiltrometer, and the infiltration time and cumulative infiltration were recorded. The detailed layout of the
test apparatus is shown in Fig. 1.
The addition of a certain amount of lactose, antifreeze or other substances to water greatly reduces the
freezing point of water (Zhao et al., 2013;Williams and Burt, 1974) so that the soil macropores are not
quickly filled with ice with decreasing temperature, thereby maintaining better conditions for water flow. To
further verify the feasibility of the use of deionized water to prepare an aqueous solution of ethylene glycol
at a mass concentration of 40% as the infiltration medium for the frozen soil measurements, the surface
tension of the aqueous glycol solution at -5°C and -10°C and its relationship with the temperature were
measured with a contact angle measuring instrument (OCA20, DataPhysics Instruments, Germany) and a
surface tension measuring instrument (DCAT-21, DataPhysics Instruments, Germany), respectively. As an
example, the contact angle measurement process of the black soil at -10°C with the aqueous ethylene glycol
solution is shown in Fig. 2, and it is observed that the contact angle decreases to 0° within a few seconds
after the liquid droplet is placed on the soil, and the liquid droplet completely dissolves in the frozen soil,
which implies that the addition of glycol to water does not alter the wetting ability of the soil particles (Lu
and Likos, 2004). The relevant physicochemical properties of the aqueous ethylene glycol solution and water
are compared in Table 2.



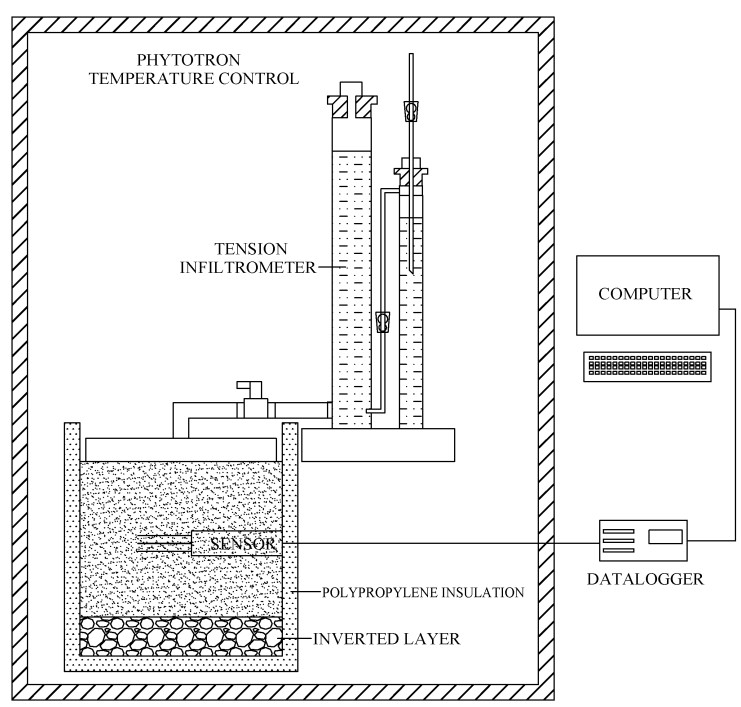


**Fig. 1**. Diagram of the test equipment.

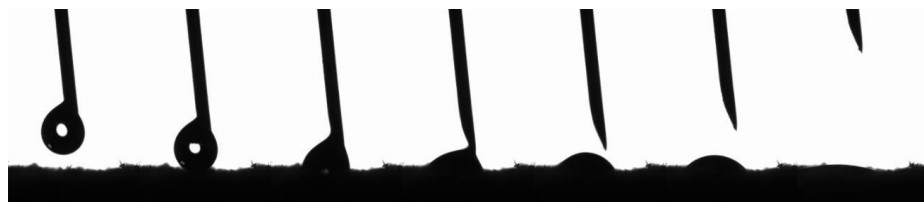


**Fig. 2**. Process of the contact angle measurement between the aqueous ethylene glycol solution and black
soil at -10°C.
**Table 2**
**Comparison of the physicochemical properties of the 40% ethylene glycol aqueous solution and**
**water**

| Infiltration solution | Temperature (°C) | Density (g/cm³) | Dynamic viscosity (mPa.s) | Surface tension (mN/m) | Contact angle (°) |
|---|---|---|---|---|---|





| | | | | | |
|---|---|---|---|---|---|
| Water | 15 | 0.9991 | 1.14 | 73.56 | 0 |
| Ethylene glycol | -5 | 1.0683 | 7.18 | 48.89 | 0 |
| aqueous solution | -10 | 1.0696 | 9.06 | 49.10 | 0 |

**2.2 Measurement of the frozen soil hydraulic conductivity**

Gardner (1958) proposed that the unsaturated hydraulic conductivity of soil varies with the matric

potential:

$$K(h) = K_{sat}\exp(\alpha h) \qquad (1)$$

where $K_{sat}$ is the saturated hydraulic conductivity, cm/hour, and h is the matric potential or tension, cm $H_2O$.

Wooding (1968) considered that the steady-state unconfined infiltration rate into soil from a circular water

source of radius R can be calculated with the following equation:

$$Q = \pi R^2 K \left[ 1 + \frac{4}{\pi R \alpha} \right] \qquad (2)$$

where Q is the amount of water entering the soil per unit time, $cm^3$/h; K is the hydraulic conductivity,

cm/hour; and $\alpha$ is a constant. Ankeny et al. (1991) proposed that implementing two successively applied

pressure heads $h_1$ and $h_2$ could yield the unsaturated hydraulic conductivity, and upon replacing K in Eq. (2)

with Eq. (1), the following is obtained:

$$Q(h_1) = \pi R^2 K_{sat} \exp(\alpha h_1) \left[ 1 + \frac{4}{\pi R \alpha} \right] \qquad (3)$$

$$Q(h_2) = \pi R^2 K_{sat} \exp(\alpha h_2) \left[ 1 + \frac{4}{\pi R \alpha} \right] \qquad (4)$$

Dividing Eq. (4) by Eq. (3) and solving for $\alpha$ yields:

$$\alpha = \frac{\ln\left[Q(h_2)/Q(h_1)\right]}{h_2 - h_1} \qquad (5)$$

where $Q(h_1)$ and $Q(h_2)$ can be measured, $h_1$ and $h_2$ are the preset tension values, and $\alpha$ can be calculated with





Eq. (5). The result can be substituted into Eq. (3) or (4) to calculate $K_{sat}$. When the number of tension levels
is larger than 2, parameter fitting methods can be applied to improve the accuracy of α and $K_{sat}$ (Hussen and
Warrick, 1993).
The tension is controlled by the bubble collecting tube of the tension infiltrometer, and different pressure
heads h correspond to different pore sizes r. By applying different pressure heads h to the soil surface, water
will overcome the surface tension in the corresponding pores and be discharged, and the infiltration volume
is recorded after reaching the stable infiltration state.
Under the assumption that the frozen soil pore ice pressure is equal to the atmospheric pressure and that
solutes are negligible, the Clausius-Clapeyron equation can be adopted to achieve the interconversion
between the soil temperature and suction (Konrad and Morgenstern, 1980;Watanabe et al., 2013), which can
be simplified as follows:
$$\psi = -L\rho_w \frac{T}{273.15} \qquad (6)$$
where ψ is the soil suction, kPa; L is the latent heat of fusion of water, $3.34 \times 10^5$ J/kg; $\rho_w$ is the density of
water, 1 g/cm³; and T is the subfreezing temperature, °C. After the unit conversion of the soil suction into h
(cm $H_2O$), the unsaturated hydraulic conductivity of frozen soil at different negative temperatures can be
obtained via substitution into Eq. (2).
**2.3 Measurement of the pore size distribution in frozen soil**
As a nonuniform medium, soil consists of pores of various pore sizes, and the equation for the soil pore
radius r can be obtained from the capillary model (Watson and Luxmoore, 1986):
$$r = -\frac{2\sigma\cos\beta}{\rho g h} \qquad (7)$$
where σ is the surface tension of the solution, g/s²; β is the contact angle between the solution and pore wall;





ρ is the density of the solution, g/cm$^3$; g is the acceleration of gravity, m/s$^2$; and h is the corresponding tension
of the tension infiltrometer, cm H$_2$O.
The effective macroporosity $\theta_m$ can be calculated for various soil particle sizes based on the Poiseuille
equation (Wilson and Luxmoore, 1988):
$\theta_m = 8\mu K_m / \rho g r^2$                    (8)
where μ is the dynamic viscosity of the fluid, g/(cm*s); $K_m$ is the macropore hydraulic conductivity and is
defined as the difference between K(h) at various tension gradients, cm/h; and r is the corresponding
equivalent pore size. The effective porosity is equal to the number of pores per unit area multiplied by the
area of the corresponding pore size. For different pore sizes, the maximum number of effective macropores
per unit area N can be calculated with the following equation:
$N = \theta_m / \pi r^2$                    (9)
where N is the number of effective macropores per unit area, and Eq. (7) calculates the minimum value  of
the pore radius, while the result obtained with Eq. (9) is actually the maximum number of effective
macropores per unit area and the maximum porosity.
Considering the differences in surface tension and density between the aqueous ethylene glycol solution and
water, when calculating the frozen soil pore size distribution, it is necessary to convert the tension into  the
equivalent pore radius according to Eq. (7), which is classified and subdivided into large and medium pores
according to the common classification method (Luxmoore, 1981), the details of which are listed in Table 3,
while the corresponding tension values in Table 3 are substituted into the fitting curve equation to calculate
the corresponding stable infiltration rate q and unsaturated hydraulic conductivity K.
**Table 3**
**Tension and equivalent pore radius conversions**





| Pore types | Pore radius (mm) | Tension conversion (cm) | | |
|---|---|---|---|---|
| | | Water (15℃) | Ethylene glycol aqueous solution (-5℃) | Ethylene glycol aqueous solution (-10℃) |
| Macroporous | >0.5 | 0~3 | 0~1.86 | 0~1.87 |
| | 0.3-0.5 | 3~5 | 1.86 ~3.11 | 1.86 ~3.12 |
| | 0.15-0.3 | 5~10 | 3.11~6.22 | 3.12~6.23 |
| Mesoporous | 0.1-0.15 | 10~15 | 6.22~9.32 | 6.23~9.35 |
| | 0.05-0.1 | 15~30 | 9.32~18.65 | 9.35~18.70 |

## 3 Results
**3.1 Infiltration characteristics of freezing-thawing soils**
According to the recorded cumulative infiltration and duration, curves of the cumulative infiltration and
infiltration rate were plotted over time, as shown in Figs. 3 and 4, respectively. The constant α and saturated
hydraulic conductivity $K_{sat}$ were calculated under different tensions h and corresponding steady-state
infiltration rates q, and the unsaturated hydraulic conductivity under different tensions was calculated with
Eq. (1). The stable infiltration rate and unsaturated hydraulic conductivity at different temperatures are
shown in Fig. 5, and the details of α and $K_{sat}$ are listed in Table 4.





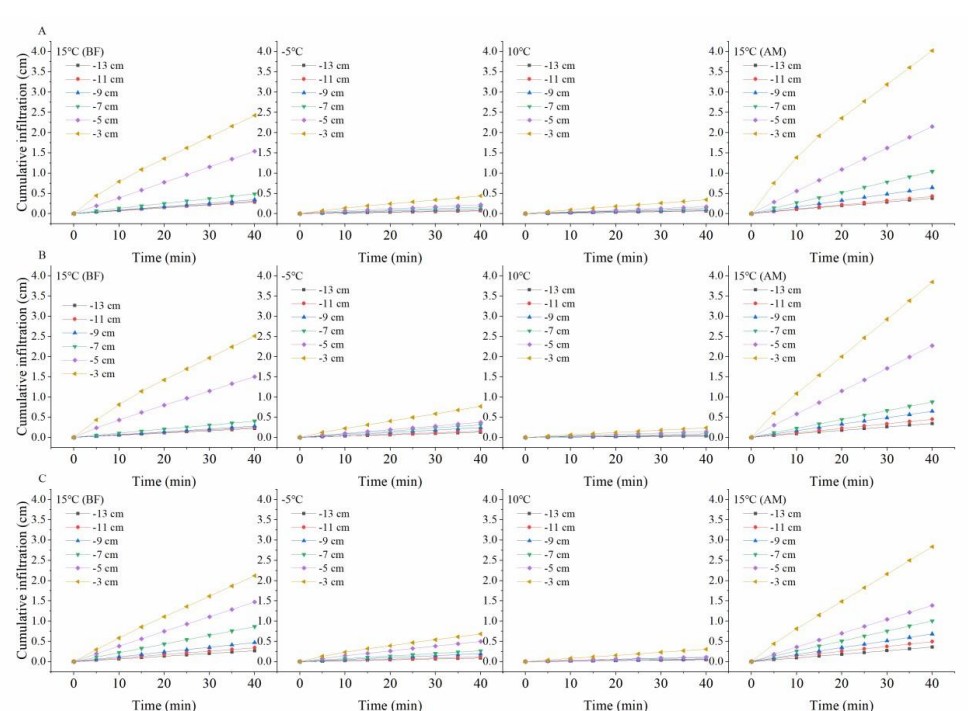

**Fig. 3.** Cumulative infiltration over time under the different treatments.

Note: A Black Soil; B Meadow Soil; C Chernozem.



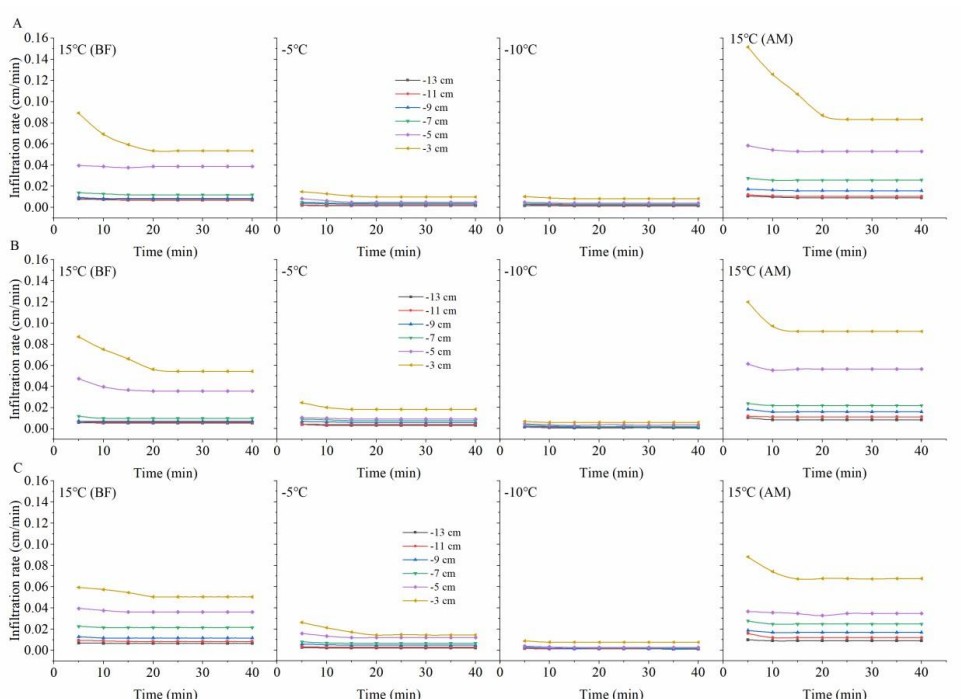


**Fig. 4**. Infiltration rate over time under the different treatments.

Note: A Black Soil; B Meadow Soil; C Chernozem.
As shown in Figs. 4 and 5, under the different tension conditions, the infiltration capacity of the unfrozen
soil is basically consistent with the findings of field experiments and is highly influenced by the tension
value (Wang et al., 1998). Compared to the room-temperature soil, the cumulative infiltration of frozen soil
slowly increases, and the infiltration rate always remains low, while under the same negative temperature
treatment, the influence of the tension value is also greatly reduced. When the temperature was reduced to -
10°C, few major tension differences were observed except for the maximum tension of -3 cm. From the
change in the slope of the two curves, we find that the time for the unfrozen soil to reach the stable infiltration
rate usually ranges from 15~20 min, while the time for the frozen soil to reach the stable infiltration rate  is
usually 10 min under higher tensions of -3 and -5 cm and 5 min under lower tensions. Comparing the
infiltration process before and after the freezing and thawing of the soil, overall, the cumulative infiltration





and infiltration rate exhibited varying degrees of increase with increasing tension value, and the increase
amplitude expanded. Moreover, the difference in the cumulative infiltration and infiltration rate between the
low tension levels ranging from -9 to -13 cm after soil thawing was larger than that before soil freezing,
which also indirectly demonstrated that freezing and thawing could further stabilize the soil pore distribution
by affecting the homogeneity, which will be detailed in subsequent sections.
**Table 4**
**Infiltration parameters of the different temperature treatments of the three soil types**

| Soil types | Temperature (°C) | $\alpha$ (cm/h) | $K_{sat}$ (cm/h) |
|---|---|---|---|
| Black soil | 15 (BF) | 0.2742 | 5.1480 |
| | -5 | 0.1993 | 0.5960 |
| | -10 | 0.2028 | 0.5221 |
| | 15 (AM) | 0.2629 | 7.4658 |
| Meadow soil | 15 (BF) | 0.3071 | 5.9232 |
| | -5 | 0.1996 | 1.1385 |
| | -10 | 0.2477 | 0.4903 |
| | 15 (AM) | 0.2934 | 9.3757 |
| Chernozem | 15 (BF) | 0.2166 | 3.7185 |
| | -5 | 0.1907 | 0.9739 |
| | -10 | 0.2508 | 0.6077 |
| | 15 (AM) | 0.2182 | 5.1283 |


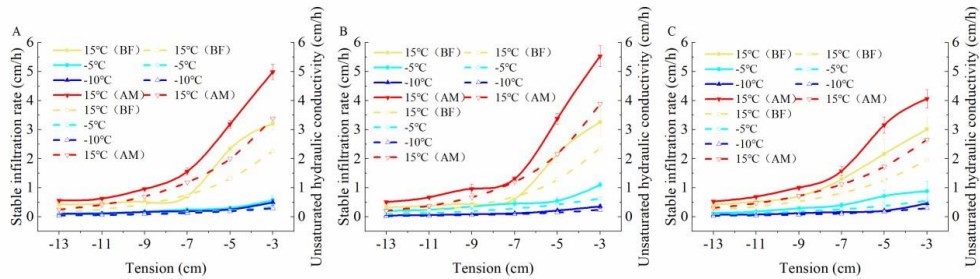






**Fig. 5**. Variation curves of the unsaturated hydraulic conductivity and stable infiltration rate with the
tension for the different treatments of the three soils.
Note: A Black soil; B meadow soil; C chernozem. The solid lines represent the stable infiltration rate, and
the dashed lines represent the unsaturated hydraulic conductivity.
Combining Fig. 5 and Table 4, we observe that the three types of soils exhibit a high infiltration capacity
under normal temperature conditions. With increasing set tension value, the suction force of the soil matrix
gradually weakens, the constraint and maintenance capacity of the matric potential to the soil water decreases,
the number of pores involved in the soil water infiltration process increases, and the unsaturated hydraulic
conductivity and stable infiltration rate of the three types of soils all reveal different degrees of increase.
When the temperature was lowered from 15°C to -5°C and the soil reached the stable frozen state, the
saturated water conductivity of the black soil, meadow soil and chernozem soil decreased by 88.42%, 80.78%
and 73.8%, respectively. With decreasing soil temperature to -10 °C, due to the presence of liquid water in
the pores, the saturated water conductivity still exhibited a certain decrease over the prefreeze conditions
and continued to decrease by 1.43%, 10.94% and 9.85%, respectively. At negative temperatures, the
unsaturated hydraulic conductivity decreased considerably and fluctuated within a small range, mainly
because the unfrozen and saturated water contents were low after soil freezing. Comparing the two
treatments of -5°C and -10°C, the unsaturated hydraulic conductivity (ANOVA, P=0.72, F=0.14) and stable
infiltration rate (ANOVA, P=0.71, F=0.15) of the black soil revealed almost no significant change, indicating
that most of its pores were filled with ice crystals at -5°C and were no longer involved in water infiltration.
The unsaturated water conductivity of the meadow and chernozem soils still exhibited a more significant
reduction when the freezing temperature was further reduced to -10°C. When the temperature was raised
again to 15°C and the soil was completely thawed, the steady infiltration rate and saturated hydraulic





conductivity increased with increasing temperature, and the values were higher than those of the soil at the
same temperature before freezing. The saturated hydraulic conductivity of the black soil, meadow soil and
chernozem increased by 45.02%, 58.63% and 37.91%, respectively, over the 15℃ (BF) treatment values.

**3.2 Pore distribution characteristics of the freezing-thawing soil**

Considering the differences in the physical and chemical properties between the infiltration solutions,
infiltration parameters such as the hydraulic conductivity and stable infiltration rate alone do not fully reflect
the infiltration characteristics and internal pore size of frozen soils. According to Eqs. (7)-(9), the maximum
number per unit area N, effective porosity $\theta_m$ and percentage of pore flow to saturated flow P corresponding
to the different soil pore sizes of the three soils under the different temperature treatments are calculated, as
shown in Figs. 6 and 7.

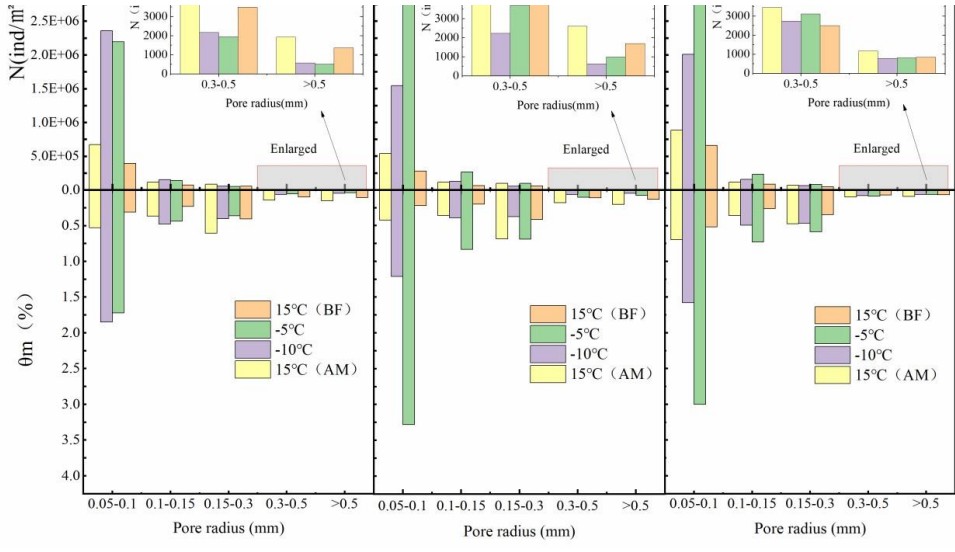

**Fig.6**. Number of pores and effective porosity of the different equivalent pores.



Note: A Black Soil; B Meadow Soil; C Chernozem.

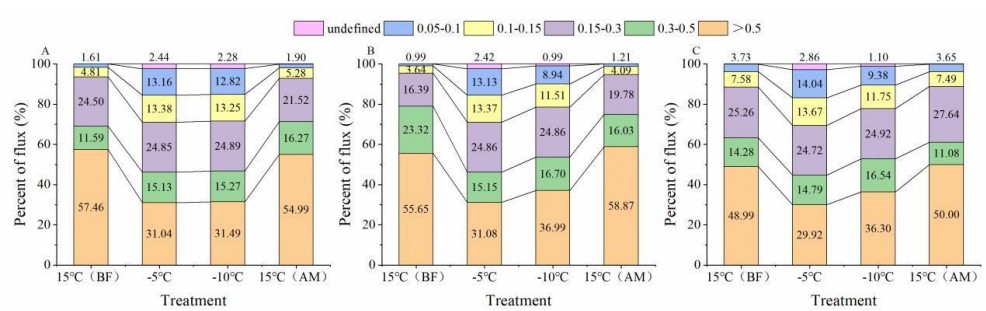


**Fig. 7**. Percentage of the pore flow in the saturated flow for the different equivalent pore sizes.
Note: A Black Soil; B Meadow Soil; CChernozem.
Fig. 6 shows that pores of different equivalent radii widely occur in all three soils, and under all four
temperature treatments, the largest N value is that for the medium pores with an equivalent radius of 0.05-
0.1 mm, and N gradually decreases with increasing equivalent radius size. Under the two room-temperature
treatments at 15°C (BF) and 15°C (AM), the largest number of 0.05- to 0.1-mm medium pores and the
smallest number of >0.5-mm macropores differed by two orders of magnitude, and the number of pores of
each size exhibited different degrees of increase or decrease over the two treatments at -5°C and -10°C where
freezing occurred, with the number of medium pores with an equivalent pore size of 0.05-0.1 mm
significantly changing. Increases of more than an order of magnitude were achieved in all three soils, while
the macropores with an equivalent pore size of >0.5 mm were generally reduced by an order of magnitude,
with the difference in the number of pores of these two sizes reaching four orders of magnitude. This
indicates that freezing caused by temperature change significantly alters the soil internal structure, with ice
crystals forming in the relatively large pores containing the internal soil moisture, resulting in a large number
of smaller pores. Assessing the two treatments at -5°C and -10°C separately, when the temperature was
lowered from -5°C to -10°C, the number of pores in each pore size interval of the meadow and chernozem



soils exhibited a significant decrease, while the black soil revealed a small increase, which might be related
to the high organic matter content of the black soil. Comparing the two treatments at 15°C (BF) and 15°C
(AM), the number of pores in all three soils increased to different degrees after thawing, and more pores
were formed with the melting of ice crystals after freeze-thaw destruction of the soil particles, which
enhanced the soil water conductivity.
Comprehensive analysis of Figs. 6 and 7 reveals that before freezing, the $\theta_m$ of the various pore sizes of the
black and meadow soils and chernozem with an equivalent radius of >0.5 mm were 0.11%, 0.13% and 0.07%,
respectively, while the P value reached 57.46%, 55.65% and 48.99%, respectively, with the values of the
thawed soil similar to these values. This indicates that for all five soil pore sizes under unfrozen conditions,
although the number of macropores with a pore size >0.5 mm is the smallest and the effective porosity is the
lowest, their contribution to the saturated flow is usually more than half, and the macropores need only
represent a small fraction of the pore volume to significantly contribute to the soil water flow. For the frozen
soil, the P value of the >0.5-mm macropores was significantly reduced and remained at approximately 30%
after the reduction, while the P value of the smaller pore sizes such as 0.15-0.3 mm, 0.1-0.15 mm, and 0.05-
0.1 mm, revealed different degrees of increase. Moreover, the smaller the pore size was, the greater the P
value increased, and their contribution eventually accounted for more than 10% of the saturated flow. The
saturated flow became more evenly distributed across the pores of each size, and the total proportion of
medium pores exceeded that of the macropores. This indicates that the freezing action caused obvious
changes to the soil structure, pore size and quantity, and although the macropores still played an  important
role, the infiltration capacity of the frozen soil no longer relied solely on these macropores, and the
contribution of certain smaller-sized mesopores to the infiltration capacity of the frozen soil could no longer
be neglected. Selecting the black soil as an example, the total effective porosity of the pores of each size



under the four treatments was 1.15%, 2.62%, 2.84%, and 1.80%, and the P values were 99.97%, 97.56%,
97.72%, and 99.96%, respectively, which implies that the soil water infiltrated almost entirely via the large
and medium pores. The small micropores, even in large numbers, contributed little to the infiltration process.

## 4 Discussion

### 4.1 Permeability and hydraulic conductivity of the frozen soil

In the field environment, although it is difficult to accurately measure the infiltration rate of frozen soils
using traditional instruments and methods such as single-loop infiltrators, the obtained test results still
demonstrate that the infiltration capacity decreases by one or more orders of magnitude when the soil is
frozen (Stähli et al., 2004). Although the cumulative infiltration and infiltration rate of frozen soil are low,
the presence of unfrozen water allows a certain amount of infiltration flow to be maintained in the soil. When
water is applied as the infiltration solution, the low temperature in the frozen soil easily causes the infiltration
water to freeze, thus forming a thin layer of ice on the soil particle surface and delaying the subsequent
infiltration of water. This phenomenon results in a low infiltration rate after the freezing of soils with a high
initial water content and a relatively high infiltration rate after the freezing of dry soils (Watanabe et al.,
2013), because the higher the ice content is, the more latent heat needs to be overcome to melt any ice crystals,
resulting in a weakened propagation of the melting front, thus limiting the infiltration rate so that it is
controlled by the downward movement of the melting leading edge of the ice crystals (Pittman et al., 2020).
During the measurements using the tension infiltrator in this study, the sensor temperature always remained
consistent with the soil temperature, indicating that the use of an aqueous glycol solution could be a useful
way to avoid the problem of freezing of the infiltration solution. In addition, the hydraulic conductivity of
frozen soils with different capacities and at various water flow rates was demonstrated not to greatly differ
(Watanabe and Osada, 2017).



Whether water or other low-freezing point solutions are applied as infiltration media, the hydraulic
conductivity of frozen soil significantly changes only within a limited temperature range above -0.5°C
depending on the unfrozen water and ice contents, and at a soil temperature below -0.5°C, the hydraulic
conductivity usually decreases to below $10^{-10}$ m/s (Watanabe and Osada, 2017;Williams and Burt, 1974).
The unsaturated hydraulic conductivity in our experiments was measured at a set tension level, and according
to Eq. (6), the soil substrate potential increases by 125 m for every 1°C decrease in temperature (Williams
and Smith, 1989), while the frozen soil hydraulic conductivity calculated at -5°C and -10°C, which
corresponds to the actual matric potential, is much lower than $10^{-10}$ m/s and can be ignored. This suggests
that even under ideal conditions where no heat exchange occurs between the infiltration solution and the soil
and no freezing of the infiltration water takes place to prevent the subsequent infiltration, the unsaturated
hydraulic conductivity of the frozen soil is so low that the frozen soil at lower temperatures in its natural
state could be considered impermeable, both for water and other solutions.
**4.2 Effect of the freeze-thaw cycles on soil pore distribution**
In our study, the N value after freezing for the different types of soil was approximately 1000-2000/m$^2$ using
the tension infiltrator, which agreed well with other studies and remained at a same magnitude (Pittman et
al., 2020), indicating that the method is generally reliable. The freeze-thaw effect significantly improves the
water conductivity of the different types of soils because it increases the porosity, decreases the soil
compactness and dry weight, and thus increases the soil water conductivity (Fouli et al., 2013). On this basis,
we also found that the freeze/thaw process significantly alters the size and number of soil pores, especially
after freezing, and the number of macropores decreases, while the contribution of macropores to the saturated
flow decreases. The proportion of the saturated flow in the mesopores with a pore size of <0.3 mm
approaches or even exceeds the proportion in the macropores, indicating that the soil water inside relatively



large pores is more likely to freeze, which in turn creates a large number of small pores, whereas the water
transfer process in unfrozen soils primarily relies on the macropores, with obvious differences (Wilson and
Luxmoore, 1988;Watson and Luxmoore, 1986). The unsaturated water conductivity of the frozen soils
measured in this study is quite low, but under human control (Watanabe and Kugisaki, 2017) or natural
conditions in the field (Espeby, 1990), water has been shown to infiltrate into frozen soils through
macropores as long as the pore size is large enough. Considering that the soil in this experiment is disturbed
soil that has been air dried and sieved, although the macropores created by tillage practices (Lipiec et al.,
2006) and invertebrate activities (Lavelle et al., 2006) are excluded, due to the inherent heterogeneity of the
soil particles, macropores remain in the uniformly filled soil column (Cortis and Berkowitz, 2004;Oswald et
al., 1997), and these macroporous pores still play a role in determining the infiltration water flow.
Studies related to the frozen soil macropore flow and pore distribution are still quite few and more data
should be acquired and more models should be developed to better understand the water movement in frozen
soil regions. In subsequent studies, we will consider applying the methods used in this paper to field
experiments to examine the dynamics of the infiltration capacity and pore distribution in nonhomogeneous
soils during whole freeze-thaw periods under real outdoor climatic conditions, such as lower temperatures
and more severe freeze-thaw cycles, but the infiltration solution must be carefully selected, as ethylene glycol
is toxic, to prevent contamination of agricultural soils and crops, and a certain concentration of lactose could
be considered (Burt and Williams, 1976;Williams and Burt, 1974). Measurements should focus on frozen
soil layers at different depths, especially in the vicinity of freezing peaks, and the spatial variability in the
distribution of frozen soil pores should be investigated. This work helps to improve the accuracy of
simulations such as those of frozen soil water and heat movement or snowmelt water infiltration processes.
**5 Conclusions**



In this paper, the infiltration capacity of soil columns under four temperature treatments representing various
freeze-thaw stages was measured, and the distribution of the pores of various sizes within the soil was
calculated based on the measurements by applying an aqueous ethylene glycol solution with a tension
infiltrator in the laboratory. The results revealed that for the three types of soils, i.e., black soil, meadow soil
and chernozem, the macropores, which accounted for only approximately 0.1% to 0.2% of the soil volume
at room temperature, contributed approximately 50% to the saturated flow, and after freezing, the proportion
of macropores decreased to 0.05% to 0.1%, while their share of the saturated flow decreased to
approximately 30%. Coupled with the even smaller mesopores, the large and medium pores, accounting for
approximately 1% to 2% of the soil volume, conducted almost all of the soil moisture under saturated
conditions. Freezing decreased the number of macropores and increased the number of smaller-sized
mesopores, thereby significantly increasing their contribution to the frozen soil infiltration capacity so that
the latter was no longer solely dependent on the macropores. The infiltration parameters and pore distribution
of the black soil were the least affected by the different negative freezing temperatures under the same
moisture content and weight capacity conditions, while those of the meadow soil were the most impacted.

## Data availability

Data used in this study are available in the Figshare (doi: 10.6084/m9.figshare.12965123 ).

## Author contributions

Ruiqi Jiang designed research program. Tianxiao Li and Ruiqi Jiang built and deployed the soil column and
instruments with assistance from Qinglin Li and Renjie Hou. Dong Liu and Qiang Fu provided funding for
test equipment. Song Cui collected soil samples in the field. Ruiqi Jiang and Tianxiao Li analyzed the
laboratory data. Ruiqi Jiang prepared the manuscript with comments from Tianxiao Li and Dong Liu.

## Competing interests



The authors declare that they have no conflict of interest

## Acknowledgements

We acknowledge that this research was supported by the National Natural Science Foundation of China
(51679039), the National Science Fund for Distinguished Young Scholars (51825901), the Heilongjiang
Provincial Science Fund for Distinguished Young Scholars (YQ2020E002),"Young Talents" Project of
Northeast Agricultural University(18QC28),China Postdoctoral Science Foundation Grant(2019M651247)

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
