# Peer review of "Soil infiltration characteristics and pore distribution under"

_The Cryosphere, 2020_

## Referee Comment (RC1) · Aaron Mohammed (Referee) · 10 Nov 2020

Summary and General Comments

The authors have presented an experimental study on the effects of freezing temperatures and one freeze-thaw cycle on soil infiltrability. Using a relatively simple laboratory experimental set-up, the authors subjected repacked soil samples to repeated tension infiltration cycles at varied temperatures and thus frozen/unfrozen states. Results showed that all samples, for the same total soil moisture (I assume, as the authors have not shown the data), exhibit a decline in unsaturated hydraulic conductivity with decreasing temperature, however only modest decreases were observed in samples once frozen. Using multiple applied tensions allowed the estimation of the maximum

pore size contributing to flow. Similar to previous work, the authors found that 'macro-pores' occupied a small portion of the total pore-volume but contributed disproportionately to flow... very nice to show experimentally that this still occurs under frozen conditions. Interestingly the authors found that the contribution of these larger pores to flow decreased when frozen, and the authors invoked soil structural changes during freezing as the mechanism to explain observations.

The paper is, for the most part, well written. Some sentences are a bit awkward, but the scientific understanding of the authors is clear and insight is still easy to comprehend. The rationale for the study and the methods are explained very well. The results are clear, however some discussion of the underlying mechanisms for the observations need clarifying and further refinement. The authors do not present some experimental data that are crucial to understanding their observations, like soil porosity, measured pre-freezing soil moisture content, and soil-frost during infiltration events. I am also very worried that the soil structure changes observed may be an artefact of the repacking of the sample before undergoing its first freezing treatment. However, the results and other aspects of the discussion are still a solid contribution, and this work should be published. However, there are some important issues that need to be addressed before I can recommend publication.

Specific Comments

An assumption underpinning the authors' tension infiltration analysis is the assumption that larger pores only flow fully saturated (no air-water interface inside the pore) and excludes the formation of an air-water interface with flowing water in larger pores. Recent work has shown this flow mode does indeed occur (see the multitude of works by Drs. John Nimmo and Peter Germann). It would be nice for the authors to acknowledge these limitations in their work.

What was the pre-freezing soil moisture content of the samples? You should show this data.

What was the porosity of the samples? If the pre-freezing volumetric moisture content was 0.3 as the authors suggest (should state more explicitly), then when frozen that will result in relatively high content if the soil porosity is say. . . 0.4 to 0.45. The authors need to clarify and discuss this.

How was the pre-freezing moisture content held consistent between samples, and after BF tests?

In unfrozen tension infiltrometer experiments, the soil moisture is assumed to be that imposed by the applied tension. If samples were frozen before infiltrometer experiments, then is it assumed that the applied tension then only affects the pores that are active during infiltration?

How was the pre-freeing water content controlled?

L51: Inappropriate reference, the review paper of Jarvis (2016) hardly mentions frozen soil dynamics, other than we do not understand it enough.

L56: 'characterization of freezing-thawing soil infiltration' sounds awkward. Do you mean infiltration into freezing/thawing soils?

L60: Daniel et al (1997) should be Stadler et al. (1997). Could also cite some other field studies on frozen soil infiltration and deeper soil percolation and refreezing effects, such as Hayashi et al (2003) and Mohammed et al. (2019).

L71-73: Zhao et al. (2013) did not introduce the 'impedance concept', it was proposed far earlier, at least as early as Jame and Norum (1980).

L75: 'results in hydraulic conductivity estimation'. . . confusing, can the authors clarify?

L89: What does the authors mean by 'freezing profiles'? Do you mean the soil freezing characteristic?

L152: Authors should state clearly that pure water was used as the infiltration solution for the unfrozen experiments. Also were the samples gravity drained after the

unfrozen test? Were the samples adjusted to ensure consistent pre-freezing soil moisture among samples?

L279: What do the authors mean by saturated water content of the frozen soil? Weren't all the samples unsaturated, so how then can there be a saturated water content?

L282-289: Was there any correlation between the amount of decrease in hydraulic conductivity from -5 °C to -10 °C and clay content or organic matter content among the soil samples?

L322-341: This is fascinating, and in my opinion, is the most novel part of this study. But this is conjecture, and there are quantitative ways to examine soil structure before and after freezing, see for example Holten et al. (2018) and Ding et al. (2019), who actually apply geophysical imaging techniques to quantity pore structure in relation to frozen soil infiltration.

L353: I agree, but you need to show those water and ice contents of the soil samples.

L368-372: Not sure I agree with this statement, as your data contradicts it, and at atmospheric pressure, air-filled macropores will conduct most water, regardless of antecedent moisture.

L376-403: I have a few issues with the discussion in this section, mostly because of a point that the authors themselves bring up... that these experiments were performed on repacked, air-dried samples. So, although they cite other studies that show that macropores may still play a role in re-packed soil samples, they did not show so in their own data. Also, the notion that macroporosity is decreased after freezing goes against other experimental studies that explicitly investigate the effect of multiple freeze-thaw cycles on soil structure (Ding et al., 2019). This may be an artifact of the fact that this was the first freeze-thaw cycle after the sample was repacked.

Additional References:

Ding, B., Rezanezhad, F., Gharedaghloo, B., Van Cappellen, P., & Passeport, E.

(2019). Bioretention cells under cold climate conditions: Effects of freezing and thawing on water infiltration, soil structure, and nutrient removal. Science of the Total Environment, 649, 749–759.

Hayashi, M., van der Kamp, G., & Schmidt, R. (2003). Focused infiltration of snowmelt water in partially frozen soil under small depressions. Journal of Hydrology, 270(3-4), 214-229.

Holten, R., Bøe, F.N., Almvik, M., Katuwal, S., Stenrød, M., Larsbo, M., Jarvis, N. & Eklo, O.M. (2018). The effect of freezing and thawing on water flow and MCPA leaching in partially frozen soil. Journal of Contaminant Hydrology, 219, 72-85.

Jame, Y.W., & D.I. Norum. (1980). Heat and mass transfer in a freezing unsaturated porous medium. Water Resources Research, 16, 811–819.

Hayashi, M., van der Kamp, G., & Schmidt, R. (2003). Focused infiltration of snowmelt water in partially frozen soil under small depressions. Journal of Hydrology, 270(3-4), 214-229.

Mohammed, A. A., Pavlovskii, I., Cey, E. E., & Hayashi, M. (2019). Effects of preferential flow on snowmelt partitioning and groundwater recharge in frozen soils. Hydrology & Earth System Sciences, 23(12).

---

## Author Comment (AC1) · 28 Nov 2020

Dear Dr. Aaron Mohammed: Thank you very much for your attention and the referee comments on our manuscript "Soil infiltration characteristics and pore distribution under freezing-thawing conditions". We are honored to have the opportunity to communicate and learn from you. If you have any other suggestions or questions after reading the responses, please feel free to contact us. Response to General Comments I'm sorry that our English is not good enough, resulting in some sentences that feel awkward to you. About the General Comments, we did ignore the potential effect of the soil porosity, so we measured the soil porosity in the column using a soil three-phase instrument in a recent supplemental test. The values of the pre-freezing soil moisture content were measured with sensors, and the values were distributed within $0.30 \pm$

0.02. The main reason for using refilling soil columns is to avoid the impact of large pores such as worm holes and plant roots; before the freezing process, all soil columns were precisely controlled and maintained at the same moisture content, so we believe there is no need to be overly concerned about the impact of this aspect. The experimental data like the soil porosity and pre-freezing soil moisture content will be added to the supplement and also to the revised manuscript if needed. We have benefited a lot from your valuable comments; the following is the responses and explanations to address your specific comments. Responses to Specific Comments As you suggest, the assumptions and limitations of large pore flow will be added in section 2.2, and we believe this will give the paper a more solid theoretical support. The following are references, and further comments are welcome if they are not comprehensive. Relevant referencesïjŽ Nimmo J R . Theory for Source-Responsive and Free-Surface Film Modeling of Unsaturated Flow [J]. Vadose Zone Journal, 2010, 9(2):295-306. Nimmo J R . Preferential flow occurs in unsaturated conditions [J]. Hydrological Processes, 2012, 26(5):786-789. Beven K , Germann P . Macropores and water flow in soils revisited [J]. Water Resources Research, 2013, 49(6):3071-3092.

The pre-freezing moisture content of the soil column was preset to 0.3, and after the column was filled, the sensor showed a moisture content distribution in the range of $0.30 \pm 0.02$. Some of the specific data from the sensors will be provided in the supplement as a reference.

The porosity of the samples was measured by a soil three-phase instrument and determined to be $0.50 \sim 0.55$. Due to the effects of the pre-freeze snowfall and snowmelt water, the water content of the field soil was relatively high, so we set the water content of the soil column at 0.3.

For each of the four temperature treatments, at least three soil columns were made for each soil type. We made over 40 soil columns in total, and the columns that were subjected to the BF tests were not subjected to other treatments. We forgot to detail that in the manuscript. We will explain this in "2.1 Test plan" in the revised manuscript.

The theoretical questions about these related assumptions have not been carefully considered by us before. May I ask for a further suggestion or note?

Each soil column was treated with only one type of temperature treatment test, so the pre-freeze water content of each soil column is approximately the same, controlled at about 0.3.

L51: Such a reference is indeed inappropriate. The paper of Jarvis (2016) only briefly describes the influence of the soil freezing and thawing in cold climates on the dynamics of the soil macroporosity and preferential flow in the Concluding Remarks. Of the papers cited by Jarvis, another review paper of Hayashi (2013) is more appropriate as a reference. Relevant references Hayashi, Masaki. The Cold Vadose Zone: Hydrological and Ecological Significance of Frozen-Soil Processes [J]. Vadose Zone Journal, 2013, 12(4):37-49.

L56: Please forgive our limited English. What I really mean is a quantitative study of the infiltration process in freezing/thawing soils.

L60: We have corrected the names of the authors of the references and read the studies on frozen soil infiltration and deeper soil percolation and refreezing effects that you recommend. These papers will be cited in the appropriate sections of the revised manuscript.

L71-73: Sorry to have confused you again. We only mean that the method used by Zhao in his article is related to the impedance coefficient; we do not consider the 'impedance concept' to have been proposed by Zhao.

L75: The problem arises from our negligence; the correct form here would be 'resulting in hydraulic conductivity overestimation'.

L89: Yes, the 'freezing profiles' here are the soil freezing characteristic curves.

L152: The infiltration solution for the unfrozen experiments we used is still deionized water; this will be stated clearly in the revised manuscript. As we have mentioned

before, all soil columns were subjected to only one temperature treatment test, so there is no need to adjust the pre-freezing soil moisture.

L279: The correct phrase here should be 'because the unfrozen water content and saturated hydraulic conductivity were low after the soil freezing'. The reason for this error is that there is only a one-word difference between the saturated water content (éěśåŠŇåŘńæřťçŐǦ) and saturated hydraulic conductivity (éěśåŠŇåŕijæřťçŐǦ) in Chinese, and writing errors lead to subsequent translation errors. We are so sorry about this mistake.

L282-289: As we stated in L317-318, we also believe that some specific experimental phenomena may be related to the higher organic matter content or clay content of black soils. However, we did not do any further research on these and nor did we find any suitable references to support this point, so we did not discuss it. If you have a better idea, please give us further suggestions.

L322-341: I also think this is the most valuable part of our study. Geophysical imaging techniques are indeed an effective method, and our department is in the process of purchasing an NMR analyzer for soil research. However, these instruments are often expensive and difficult to carry around. The tension infiltrometer used in this article is affordable and widely used, and we believe that this article can provide an important reference for its use in winter field tests.

L353: We did not measure soil ice content in our previous experiments because our neutron meter was banned and recycled by the local environmental agency. In this supplemental experiment, we used an electric drill to collect soil samples and then dried them, but due to the soil disturbance caused by the drill, I personally believe that the data is only suitable as a reference. These data will be provided in the supplement.

L368-372: Despite the fact that air-filled macropores will conduct most of the water, the freezing of the soil moisture could considerably change the arrangement and bonding of the soil particles and thus change the soil structure (Bullock et al. 2001). Freezing

and thawing could also lead to the mechanical fragmentation of coarse soil particles and the aggregation of fine soil particles (Zhang et al. 2016). Therefore, the pore connectivity and hydraulic conductivity of freezing and thawing soils will also be affected. Relevant references Bullock M S , Larney F J , R.César Izaurralde, et al. Overwinter Changes in Wind Erodibility of Clay Loam Soils in Southern Alberta[J]. Soil ence Society of America Journal, 2001, 65(2):423-430. Ze Z , Wei M A , Wenjie F , et al. Reconstruction of Soil Particle Composition During Freeze-Thaw Cycling: A Review[J]. Pedosphere 26:167–179.

L376-403: We believe that the data in Figures 6 and 7 about the number of pores, the effective porosity, and the percentage of the pore flow in the saturated flow have been able to show that macropores may still play a role in re-packed soil samples. The macroporosity did decrease after freezing, but the thawed soil had a higher porosity and a greater number of pores of different sizes compared to unfrozen soil, which does not conflict with the study (Ding et al., 2019) whose conclusion is 'FTCs resulted in larger pores and more small pores maintaining high infiltration'. In addition, the research methods used in this article are significantly different from ours, for example, prior to the start of each FTC, the injection solution was added from the top into the soil column to the point of saturation. Indeed, just as you suggest, it is possible that our conclusions may be an artifact of the fact that this was the first freeze-thaw cycle after the sample was repacked, so this speculation will be added to the discussion.

Please also note the supplement to this comment:
https://tc.copernicus.org/preprints/tc-2020-280/tc-2020-280-AC1-supplement.zip
* * *

---

## Referee Comment (RC2) · Anonymous Referee #2 · 8 Jan 2021

The manuscript describes experiments of infiltration into frozen and thawed soils, and calculations of pore distributions and their changes with freezing. Laboratory experiments are carried out on three soil types and infiltration of a glycol solution. The results should be of interest for researcher interested in hydrology in areas that undergo seasonal freezing or have permafrost, and perhaps in particular modelers who intend to represent these processes. Especially the pore size distribution results seem novel. The manuscript is generally well-written but could need a checking of the language as some sentences are long and not easy to follow. I recommend major revisions and that the authors take another look at the language and structure so that the paper is easier to read after revision.

Major comments

[Figure]

My main concern about the manuscript regards how the results can be related to infiltration of water in soil, as the solution used here has different properties from water (e.g. viscosity). Are the presented values of estimated hydraulic conductivity for the glycol solution or for water? It would be most helpful to present values for water, or perhaps permeability values rather than hydraulic conductivity values.

Clarification of the water content of samples is needed to understand how these freezing processes can be related to freezing of soil in field conditions. Is there water in the samples before the solution is added, and if so how much? Is it the water (which was already in the soil before addition of solution) that freezes in the soil pores or is it the added solution that partly freezes?

Clarifying the two points above would make the manuscript much more valuable for scientists looking to relate these laboratory experiments to field conditions.

Minor comments

L20-24: The sentence seems incomplete. L26: replace first comma with "and" Throughout, insert space after semicolon when several references are listed within a parenthesis Table 1. What are the soil textures for meadow and chernozem soils? L188-189: Remove subscript format from reference.

Methods: What was the initial water content of samples? Was water or the aqueous solution used for the experiments, or both? How much of the liquid was frozen?

Big difference in viscosity for water and the aqueous solution. So conductivity is for this solution and not for water – should be converted to water?

Figures 3 and 4 – are both needed? Don't they more or less show the same thing?

L45: Do you mean figures 3 and 4?

Figure 5 would benefit from a more detailed description and discussion in text. There is a lot of information in this figure and I cannot distinguish 12 separate lines in each plot.

[Figure]

What is really unsaturated hydraulic conductivity and why is this included? Hydraulic conductivity should vary with saturation, but is there a fixed level of saturation and if so, what saturation level is this?

L273: What is meant by "stable frozen"? Is all water/liquid turned to ice at this temperature?

Fig 6: check Y axis title Fig 6: is there any uncertainty related to these estimates?

---

## Author Comment (AC2) · 14 Jan 2021

Response to the referee comments by Anonymous Referee #2

Dear Referee: Thank you very much for your attention and the referee comments on our manuscript "Soil infiltration characteristics and pore distribution under freezing-thawing conditions". We are honored to have the opportunity to communicate with you and learn from you; if you have any other suggestions or questions after reading the responses, please feel free to contact us.

Response to Major Comments

We apologize that our English is not good enough, resulting in some long sentences that may be awkward to read. The revised paper will be polished by professional or-

ganizations. The estimated hydraulic conductivity of frozen soil is considered for the glycol solution, and the hydraulic conductivity of unfrozen soil is considered for the water. The saturated hydraulic conductivity can reflect the permeability of the soil to some extent, and the values of saturated hydraulic conductivity are given in Table 4. The water content of the soil sample is given in the main text, as seen in L136. The pre-freezing water content of the soil column was preset to 0.3, and after the column was filled, the sensor showed a moisture content in the range of $0.30 \pm 0.02$. Some of the specific data from the sensors will be provided in the supplement as a reference.

Response to Minor Comments

L20-24: As you suggest, the statement here is indeed incomplete. The complete expression should be 'selected black and meadow soils and chernozem as test subjects.' We will take another look at the language and structure of the manuscript after revision have been made.

L26 and L188-189: Punctuation and formatting issues will be corrected in the revised version.

Table 1: We have identified the cause of the format conversion problem during submission. The soil textures of the meadow and chernozem soils are both silt loam.

Methods: The initial water content of the samples was 0.3, and water and aqueous solutions were both used for the experiments. In a supplemental experiment, we used an electric drill to collect soil samples and then dried them. The unfrozen water content and ice content data of the soil samples are provided in the supplement. The hydraulic conductivity in frozen soil is for glycol solution, and that in unfrozen soil is determined for water.

Figures 3 and 4 do have similarities, but cumulative infiltration and infiltration rate are different concepts, and we believe they provide a more complete reflection of the changes in soil infiltration capacity.

L45: Figure 4 represents the infiltration rate over time under the different treatments. The unsaturated hydraulic conductivity is shown in Figure 5, and the saturated hydraulic conductivity is given in Table 4.

Figure 5: In the revised paper, we will provide a more detailed description and discussion in the text. The saturation of the soil changed significantly after freezing, and the saturated hydraulic conductivity better reflected the relevant issues.

L273: The stable frozen state usually indicates that no drastic changes in temperature and water content occur.

Fig. 6: The Y-axis of the internal expansion chart will be standardized to scientific notation.

Please also note the supplement to this comment:
https://tc.copernicus.org/preprints/tc-2020-280/tc-2020-280-AC2-supplement.zip

---

## Author Response (AR1)

**Reply to Review Comments**

Dear Editors:

Thank you for seeing our effort and helping us further improve the manuscript, "Soil infiltration characteristics and pore distribution under freezing-thawing conditions", which was invited for revision in The Cryosphere. We really appreciate your thoughtful comments. We have taken these comments to heart and substantially improved our manuscript in response to the review comments we received.

**In summary:**

1) We acknowledged the limitations of the relevant fundamental theory and materials.
2) We added the data of soil porosity and pre-freezing moisture content of samples.
3) We showed the unfrozen water content and ice content of the frozen samples.
4) We revised formatting and detail mistakes based on comments of reviewers.
5) We conducted a detailed self-review, and the revised manuscript was re-polished by a professional institution.

Below, please find a detailed set of responses to specific comments. We referred to the most related lines in the unmarked version of the manuscript. If there is anything else you feel needs to be revised, please feel free to contact us.

**Comments from Referee #1: Mohammed, Aaron**

**Comments:** An assumption underpinning the authors' tension infiltration analysis is the assumption that larger pores only flow fully saturated (no air-water interface inside the pore) and excludes the formation of an air-water interface with flowing water in larger pores. Recent work has shown this flow mode does indeed occur (see the

multitude of works by Drs. John Nimmo and Peter Germann). It would be nice for the authors to acknowledge these limitations in their work.

**Response:** We added the assumptions and limitations of large pore flow in Discussion, which can be seen in L355-359.

**Comments:** What was the pre-freezing soil moisture content of the samples? You should show this data.

**Response:** These data are now presented in Table 2, in L141.

**Comments:** What was the porosity of the samples? If the pre-freezing volumetric moisture content was 0.3 as the authors suggest (should state more explicitly), then when frozen that will result in relatively high content if the soil porosity is say…0.4 to 0.45. The authors need to clarify and discuss this.

**Response:** The data of soil porosity were also placed in Table 2.

**Comments:** How was the pre-freezing moisture content held consistent between samples, and after BF tests?

**Response:** For each of the four temperature treatments, at least three soil columns were made for each soil type. We made over 40 soil columns in total, and the columns that were subjected to the BF tests were not subjected to other treatments. We explained this in L131.

**Comments:** In unfrozen tension infiltrometer experiments, the soil moisture is assumed to be that imposed by the applied tension. If samples were frozen before infiltrometer experiments, then is it assumed that the applied tension then only affects the pores that are active during infiltration?

**Response:** As shown in Table 2, the water content of the soil had been unified to about 0.30 before the infiltration experiments. We believe that after the freezing of the soil, the infiltration process is mainly influenced by the pores that have not been blocked by ice crystals.

**Comments:** How was the pre-freeing water content controlled?

**Response:** Each soil column was treated with only one type of temperature treatment test, so the pre-freeze water content of each soil column is approximately the same, controlled at about 0.3.

**Comments:** L51: Inappropriate reference, the review paper of Jarvis (2016) hardly mentions frozen soil dynamics, other than we do not understand it enough.

**Response:** Another review paper of Hayashi (2013) was cited here as a reference.

**Comments:** L56: 'characterization of freezing-thawing soil infiltration' sounds awkward. Do you mean infiltration into freezing/thawing soils?

**Response:** It has been changed to 'quantitative studies of the infiltration process in freezing-thawing soils.'

**Comments:** L60: Daniel et al (1997) should be Stadler et al. (1997). Could also cite some other field studies on frozen soil infiltration and deeper soil percolation and refreezing effects, such as Hayashi et al (2003) and Mohammed et al. (2019).

**Response:** We corrected the author's name of the reference and cited the recommended papers, it can be seen in L59-62.

**Comments:** L71-73: Zhao et al. (2013) did not introduce the 'impedance concept', it was proposed far earlier, at least as early as Jame and Norum (1980).

**Response:** Sorry to have confused you again. We only mean that the method used by Zhao in his article is related to the impedance coefficient; we do not consider the 'impedance concept' to have been proposed by Zhao.

**Comments:** L75: 'results in hydraulic conductivity estimation'… confusing, can the authors clarify?

**Response:** It has been changed to 'results in an overestimation of hydraulic conductivity.'

**Comments:** L89: What does the authors mean by 'freezing profiles'? Do you mean the soil freezing characteristic?

**Response:** Yes, the 'freezing profiles' has been changed to the 'soil freezing characteristic curves.'

**Comments:** L152: Authors should state clearly that pure water was used as the infiltration solution for the unfrozen experiments. Also were the samples gravity drained after the unfrozen test? Were the samples adjusted to ensure consistent pre-freezing soil moisture among samples?

**Response:** The infiltration solution for the unfrozen experiments we used is still deionized water; this was stated clearly, as can be seen in L157. As we have mentioned before, all soil columns were subjected to only one temperature treatment test, so there is no need to adjust the pre-freezing soil moisture.

**Comments:** L279: What do the authors mean by saturated water content of the frozen soil? Weren't all the samples unsaturated, so how then can there be a saturated water content?

**Response:** Here has been revised to 'because the unfrozen water content and saturated hydraulic conductivity were low after the soil freezing', now it can be seen in L289.

**Comments:** L282-289: Was there any correlation between the amount of decrease in hydraulic conductivity from -5°C to -10°C and clay content or organic matter content among the soil samples?

**Response:** As we stated in L327-328, we also believe that some specific experimental phenomena may be related to the higher organic matter content or clay content of black soils. However, we did not do any further research on these and nor did we find any suitable references to support this point, so we did not discuss it.

**Comments:** L322-341: This is fascinating, and in my opinion, is the most novel part of

this study. But this is conjecture, and there are quantitative ways to examine soil structure before and after freezing, see for example Holten et al. (2018) and Ding et al. (2019), who actually apply geophysical imaging techniques to quantity pore structure in relation to frozen soil infiltration.

**Response:** I also think this is the most valuable part of our study. Geophysical imaging techniques are indeed an effective method, and our department is in the process of purchasing an NMR analyzer for soil research. However, these instruments are often expensive and difficult to carry around. The tension infiltrometer used in this article is affordable and widely used, and we believe that this article can provide an important reference for its use in winter field tests.

**Comments:** L353: I agree, but you need to show those water and ice contents of the soil samples.

**Response:** Unfrozen water contents and ice contents of the soil samples were listed in Table 5, L245.

**Comments:** L368-372: Not sure I agree with this statement, as your data contradicts it, and at atmospheric pressure, air-filled macropores will conduct most water, regardless of antecedent moisture.

**Response:** Despite the fact that air-filled macropores will conduct most of the water, the freezing of the soil moisture could considerably change the arrangement and bonding of the soil particles and thus change the soil structure (Bullock et al. 2001). Freezing and thawing could also lead to the mechanical fragmentation of coarse soil particles and the aggregation of fine soil particles (Zhang et al. 2016). Therefore, the pore connectivity and hydraulic conductivity of freezing and thawing soils will also be affected.

Relevant references

Bullock M S , Larney F J , R.César Izaurralde, et al. Overwinter Changes in Wind Erodibility of Clay Loam Soils in Southern Alberta[J]. Soil ence Society of America Journal, 2001, 65(2):423-430. Ze Z , Wei M A , Wenjie F , et al. Reconstruction of Soil

Particle Composition During Freeze-Thaw Cycling: A Review[J]. Pedosphere 26:167–179.

**Comments:** L376-403: I have a few issues with the discussion in this section, mostly because of a point that the authors themselves bring up…that these experiments were performed on repacked, air-dried samples. So, although they cite other studies that show that macropores may still play a role in re-packed soil samples, they did not show so in their own data. Also, the notion that macroporosity is decreased after freezing goes against other experimental studies that explicitly investigate the effect of multiple freeze-thaw cycles on soil structure (Ding et al., 2019). This may be an artifact of the fact that this was the first freeze-thaw cycle after the sample was repacked.

**Response:** We believe that the data in Figures 6 and 7 about the number of pores, the effective porosity, and the percentage of the pore flow in the saturated flow have been able to show that macropores may still play a role in re-packed soil samples. The macroporosity did decrease after freezing, but the thawed soil had a higher porosity and a greater number of pores of different sizes compared to unfrozen soil, which does not conflict with the study (Ding et al., 2019) whose conclusion is 'FTCs resulted in larger pores and more small pores maintaining high infiltration'. In addition, the research methods used in this article are significantly different from ours, for example, prior to the start of each FTC, the injection solution was added from the top into the soil column to the point of saturation. Indeed, just as you suggest, it is possible that our conclusions may be an artifact of the fact that this was the first freeze-thaw cycle after the sample was repacked, so this speculation was added to the discussion, can be seen in L411-412.

**Comments from Anonymous Referee #2**

**Comments:** My main concern about the manuscript regards how the results can be related to infiltration of water in soil, as the solution used here has different properties from water (e.g. viscosity). Are the presented values of estimated hydraulic

conductivity for the glycol solution or for water? It would be most helpful to present values for water, or perhaps permeability values rather than hydraulic conductivity values.

**Response:** The estimated hydraulic conductivity of frozen soil is considered for the glycol solution, and the hydraulic conductivity of unfrozen soil is considered for the water. The saturated hydraulic conductivity can reflect the permeability of the soil to some extent, and the values of saturated hydraulic conductivity are given in Table 4. Values for water of frozen samples as shown in Table 5. In addition, the use of ethylene glycol aqueous solution in frozen soil infiltration can minimize the effect of ice crystal erosion, except for comparison with the hydraulic conductivity of unfrozen soils, the main use of these unsaturated hydraulic conductivities is to calculate the soil pore distribution. The use of ethylene glycol aqueous solution is feasible in the experimental method, but if applied to real soil water model requires subsequent in-depth study, in the discussion section we added about the limitations, can be seen in L418-421. This suggestion will be the direction of our future efforts, we believe that a field experiment or an in-situ soils for indoor freezing tests would be more useful to research the relevant issues.

**Comments:** Clarification of the water content of samples is needed to understand how these freezing processes can be related to freezing of soil in field conditions. Is there water in the samples before the solution is added, and if so how much? Is it the water (which was already in the soil before addition of solution) that freezes in the soil pores or is it the added solution that partly freezes?

**Response:** The water content of the soil sample is given in the main text, as seen in L136. The pre-freezing water content of the soil column was preset to 0.3, and after the column was filled, the sensor showed a moisture content in the range of $0.30 \pm 0.02$. These data are now presented in Table 2

**Comments:** L20-24: The sentence seems incomplete.

**Response:** As you suggest, the statement here is indeed incomplete. The expression

was changed to 'black soils, meadow soils and chernozem were selected as test subjects.'

**Comments:** L26: Replace first comma with "and". Throughout, insert space after semicolon when several references are listed within a parenthesis.

**Response:** Punctuation issues has been corrected.

**Comments:** Table 1. What are the soil textures for meadow and chernozem soils?

**Response:** We have identified the cause of the format conversion problem during submission. The soil textures of the meadow and chernozem soils are both silt loam. Now, they have been correctly presented.

**Comments:** L188-189: Remove subscript format from reference.

**Response:** Formatting issues has been corrected.

**Comments:** Methods: What was the initial water content of samples? Was water or the aqueous solution used for the experiments, or both? How much of the liquid was frozen?

**Response:** The initial water content of the samples was 0.3, and water and aqueous solutions were both used for the experiments. In a supplemental experiment, we used an electric drill to collect soil samples and then dried them. Unfrozen water contents and ice contents of the soil samples were listed in Table 5, L245.

**Comments:** Big difference in viscosity for water and the aqueous solution. So conductivity is for this solution and not for water – should be converted to water?

**Response:** The hydraulic conductivity in frozen soil is for glycol solution, and that in unfrozen soil is determined for water.

**Comments:** Figures 3 and 4 – are both needed? Don't they more or less show the same thing?

**Response:** Figures 3 and 4 do have similarities, but cumulative infiltration and infiltration rate are different concepts, and we believe they provide a more complete

reflection of the changes in soil infiltration capacity.

**Comments:** L45: Do you mean figures 3 and 4?

**Response:** Figure 4 represents the infiltration rate over time under the different treatments. The unsaturated hydraulic conductivity is shown in Figure 5, and the saturated hydraulic conductivity is given in Table 6.

**Comments:** Figure 5 would benefit from a more detailed description and discussion in text. There is a lot of information in this figure and I cannot distinguish 12 separate lines in each plot. What is really unsaturated hydraulic conductivity and why is this included? Hydraulic conductivity should vary with saturation, but is there a fixed level of saturation and if so, what saturation level is this?

**Response:** Figure 5 has been modified, the previous legend had an error, there are only 8 separate lines in each plot. The saturation of the soil changed significantly after freezing, and the saturated hydraulic conductivity better reflected the relevant issues.

**Comments:** L273: What is meant by "stable frozen"? Is all water/liquid turned to ice at this temperature?

**Response:** The stable frozen state usually indicates that no drastic changes in temperature and water content occur.

**Comments:** Check Y axis title Fig 6: is there any uncertainty related to these estimates?

**Response:** The Y-axis of the internal expansion chart was standardized to scientific notation.

---

## Author Response (AR2)

**Reply to Editor Comments**

Dear Editors:

Thank you very much for your further guidance on our manuscript within your busy schedule, and we have revised the article in more detail based on your and the referee's comments.

**In summary:**

1) We explained the kinds of solutions used for the different experimental treatments.
2) We added a comparison with references recommended by referees.
3) We acknowledged the effect of the solution viscosities on the infiltration parameters.
4) We added the value of the permeability in Table 6.
5) We discussed the potential uncertainty in the Y-axis of Figure 6.

Below, please find a detailed set of responses to specific comments. We referred to the most related lines from the latest unmarked revision of the manuscript. If there are still anything else you feel needs to be revised, please feel free to contact us.

**Comments from Referee #1:**

**Comments #4:** How was the pre-freezing moisture content held consistent between samples, and after BF tests?

**Comments #6:** How was the pre-freeing water content controlled?

**Response:** The use of a spray bottle to add water in small quantities but several times,

and the use of sensors to monitor the moisture values guarantee the uniformity of the moisture content of the soil columns and the relative consistency of the pre-freeze moisture between different columns. This part is shown in L136-140.

**Comments #5:** In unfrozen tension infiltrometer experiments, the soil moisture is assumed to be that imposed by the applied tension. If samples were frozen before infiltrometer experiments, then is it assumed that the applied tension then only affects the pores that are active during infiltration?

**Response:** We have acknowledged the relevant assumptions and limitations in the discussion section as suggested by the referee, which will make the theoretical support of the article more rigorous. As shown in L366-368.

**Comments #10:** L71-73: Zhao et al. (2013) did not introduce the 'impedance concept', it was proposed far earlier, at least as early as Jame and Norum (1980).

**Response:** We have added the reference suggested by the referee in the corresponding places. As shown in L74.

Relevant references

Jame, Y.-W., D.I. Norum. Heat and mass transfer in a freezing unsaturated porous medium[J]. Water Resources Research, 1980, 16(4):811–819.

**Comments #13:** L152: Authors should state clearly that pure water was used as the infiltration solution for the unfrozen experiments. Also were the samples gravity drained after the unfrozen test? Were the samples adjusted to ensure consistent pre-freezing soil moisture among samples?

**Response:** This was our oversight, there is really no clear statement in the previous manuscript. We have added a clear explanation in the test plan section, which can be seen in L170-172. In addition, Table 3 shows the physicochemical properties of two

liquids, water (15℃) and ethylene glycol aqueous solution(-5℃and-10℃).

**Comments #16:** L322-341: This is fascinating, and in my opinion, is the most novel part of this study. But this is conjecture, and there are quantitative ways to examine soil structure before and after freezing, see for example Holten et al. (2018) and Ding et al. (2019), who actually apply geophysical imaging techniques to quantity pore structure in relation to frozen soil infiltration.

**Response:** References mentioned by the reviewers are added to the discussion section with appropriate comparative comments and can be seen in L404-411.

**Comments #18:** L368-372: Not sure I agree with this statement, as your data contradicts it, and at atmospheric pressure, air-filled macropores will conduct most water, regardless of antecedent moisture.

**Response:** The values here are calculated using Equation 6, as they are very small and close to the zero point of the unsaturated hydraulic conductivity variation curves in Figure 5. We gave a brief description in the Discussion section and acknowledged the effect of different fluid viscosities on the infiltration parameters, as can be seen in L393-394 and L398-401.

**Comments from Referee #2:**

**Comments #1:** My main concern about the manuscript regards how the results can be related to infiltration of water in soil, as the solution used here has different properties from water (e.g. viscosity). Are the presented values of estimated hydraulic conductivity for the glycol solution or for water? It would be most helpful to present values for water, or perhaps permeability values rather than hydraulic conductivity values.

**Response:** We added the value of permeability to Table 6 based on your and the

referee's comments, but there is a slight difference between our calculated value and the one you gave, your value is about 1.08 times of our result. We have used the following formula to calculate the permeability, but we are not sure if the formula or other factors are responsible for these differences. If you find the error in it, please give us further guidance.

$$Permeability = \frac{K_{sat}\mu}{\rho g}$$

where $K_{sat}$ is the saturated hydraulic conductivity, m/s; $\mu$ is the dynamic viscosity of the fluid, mPa·s;

$\rho$ is the density of liquid, $kg/m^3$; g is the acceleration of gravity, 9.8N/kg.

**Comments #12**: L273: What is meant by "stable frozen"? Is all water/liquid turned to ice at this temperature?

**Response:** The stable frozen state usually indicates that no drastic changes in temperature and water content occur. It has been clarified in text, as shown in L290-291.

**Comments:** Check Y axis title Fig 6: is there any uncertainty related to these estimates?

**Response:** We are not quite sure what exactly this uncertainty means, so in the previous revision we just standardized the Y-axis of the internal expansion chart to scientific notation. For the Y-axis of Fig. 6, the uncertainty of $N$ and $\theta_m$ values mainly comes from the soil pore radius $r$, and we expanded a related discussion, as seen in L427-431.